# Dietary-Fibre-Rich Fractions Isolated from Broccoli Stalks as a Potential Functional Ingredient with Phenolic Compounds and Glucosinolates

**DOI:** 10.3390/ijms232113309

**Published:** 2022-11-01

**Authors:** Vanesa Núñez-Gómez, Rocío González-Barrio, Nieves Baenas, Diego A. Moreno, Mª Jesús Periago

**Affiliations:** 1Department of Food Technology, Food Science and Nutrition, Faculty of Veterinary Sciences, Regional Campus of International Excellence “Campus Mare Nostrum”, Biomedical Research Institute of Murcia (IMIB-Arrixaca-UMU), University of Murcia, Espinardo, 30100 Murcia, Spain; 2Phytochemistry and Healthy Food Lab, Department of Food Science and Technology, Centro de Edafología y Biología Aplicada del Segura (CEBAS), CSIC, Campus Universitario de Espinardo, Edificio 25, 30100 Murcia, Spain

**Keywords:** *Brassica oleracea*, by-products, (poly)phenols, glucosinolates, prebiotic effect, SCFAs

## Abstract

The *Brassica oleracea* industry generates large amounts of by-products to which value could be added because of the characteristics of their composition. The aim was to extract different fibre fractions from broccoli stalks to obtain potential new added-value ingredients. Using an ethanol and water extraction procedure, two fibre-rich fractions (total fibre fraction, TF_B_, and insoluble fibre fraction, IF_B_) were obtained. These fractions were analysed to determine the nutritional, (poly)phenols and glucosinolates composition and physicochemical properties, comparing the results with those of freeze-dried broccoli stalks (DBS). Although TF_B_ showed a higher content of total dietary fibre, IF_B_ had the same content of insoluble dietary fibre as TF_B_ (54%), better hydration properties, higher content of glucosinolates (100 mg/100 g d.w.) and (poly)phenols (74.7 mg/100 g d.w.). The prebiotic effect was evaluated in IF_B_ and compared with DBS by in vitro fermentation with human faecal slurries. After 48 h, the short-chain fatty acid (SCFA) production was higher with IF_B_ than with DBS because of the greater presence of both uronic acids, the main component of pectin, and (poly)phenols. These results reveal that novel fibre-rich ingredients—with antioxidant, technological and physiological effects—could be obtained from broccoli stalks by using green extraction methods.

## 1. Introduction

Broccoli (*Brassica oleracea*) production and consumption have increased in the last decade because of changes in population lifestyles leading to a high adherence to healthy diets [1]. The region of Murcia (south-eastern Spain) is the major production area in Europe [2], and here the waste from the broccoli industry (leaves and stalks) has become an environmental problem for the agri-food industries. In this regard, only 15% of the total plant biomass is constituted by the florets (the edible part), and a large quantity of by-products is generated after harvesting. The other aerial parts of the plants are the stalks and leaves, which represent approximately 21% and 47%, respectively, whereas the roots represent 17% [3].

Broccoli is a plant food with a significant content of (poly)phenols and glucosinolates, bioactive compounds that can be found also in its by-products [1,3]. In this sense, broccoli stalks are rich in DF, also containing significant amounts of some minerals (such as calcium and iron), bioactive compounds (glucosinolates, (poly)phenols and carotenoids) and some vitamins (such as C and A) but in smaller quantities than in florets and leaves [3]. It has been reported that dietary fibre may affect the bioavailability of minerals and phenolic compounds, since they are bound to the fibre polysaccharides. However, when they reach the colon, where they are released by the action of the microbiota, they could be absorbed, counteracting this effect [4,5]. (Poly)phenols occur in plants in their free forms and can therefore be extracted with aqueous-organic solvents (extractable (poly)phenols (EPP)), but they are also bound to plant cell wall compounds as non-extractable (poly)phenols (NEPP), so these bonds need to be broken before their extraction. It is important to take into account the NEPP, as they make up more than 50% of the total (poly)phenols, but in most research work they have not been considered [6,7]. Particularly, in samples with a high content of DF, as is the case of stalks, the extraction of (poly)phenolic compounds with methanol:water underestimated the content of these bioactive compounds. There are several studies about the use of plant by-products for the extraction of DF using different methods [8,9], but the extraction of fibre from broccoli by-products has been scarcely investigated. A recent study has described the composition of DF obtained from broccoli by-products (a mix of florets, leaves and stalks) using different extraction methods, but the authors did not investigate the content of antioxidant bioactive compounds that remained in the isolated DF [10].

Taking into consideration their composition, stalks represent a source of DF that can be extracted from the by-products, including also the (poly)phenols. The consumption of DF has several beneficial effects on human health and DF is also considered as a prebiotic substance. The non-starch polysaccharides of DF can be fermented by the gut microbiota, producing short chain fatty acids (SCFAs) [11,12], such as acetic, propionic and butyric acids. Subsequently, SCFAs have been reported to play an important role in the maintenance of host metabolic and intestinal health. SCFAs contribute to maintaining the integrity of the intestine, regulating luminal pH by decreasing the presence of pathogenic species, increasing apoptosis and preventing cancer proliferation, regulating mucus production and acting as an energy substrate for intestinal epithelial cells and have effects on mucosal immune function [13,14,15]. After their absorption in the large intestine and distribution through the bloodstream, they modulate several functions, such as the immune response, appetite regulation, glucose homeostasis and lipid metabolism, as well as regulate the body weight and reduce the risk of some types of cancer [13,16].

Although these by-products are sometimes used for animal feeding, in many cases they are left in the field, becoming an environmental problem. Nowadays, there is greater interest in reducing the disposal (without further use) of the by-products, implementing a transition to a circular economy process in which new products with added value are designed to give a second useful life to by-products [17]. Therefore, the by-products from the agri-food industry can be processed for the extraction of constituents such as dietary fibre (DF), bioactive compounds and enzymes, the aim being to valorise them and create new production chains. In this regard, in the isolation of a DF-rich constituent, the methods used for DF extraction are very important. Extraction methods using solvents or enzymes have been commonly employed, leading to more expensive processes, as well as toxicity and environmental problems because of the toxicity of the solvents used. As an alternative, extraction techniques using microwaves, ultrasounds or supercritical fluids have emerged, but they require a large amount of energy, as well as specific equipment [18,19]. Therefore, the development of new extraction techniques that are industrially simple and environmentally friendly is very important for the implementation of the circular economy system [20].

Against this background, the aim of this work was to determine the nutritional composition of the broccoli stalk and to isolate different DF fractions using environmentally friendly extraction methodologies to valorise this by-product by obtaining new potential ingredients. The different samples obtained were analysed to characterise their proximate composition, the content of non-starch polysaccharides and the bioactive compounds profile, as well as their physicochemical properties. In addition, the potential prebiotic effect was also evaluated.

## 2. Results and Discussion

The extraction methods applied to obtain the total fibre fraction (TF_B_) and insoluble fibre fraction (IF_B_) achieved a yield of 67% and 70% from fresh broccoli stalks, respectively. These methods used ethanol-water (80:20) and water as environmentally friendly solvents to obtain the TF_B_ and IF_B_ fractions, respectively. Hence, broccoli stalk waste can be considered an interesting plant material for the extraction of fractions rich in total and insoluble DF for use as potential added-value ingredients. The nutritional composition of both fractions was studied and compared with that of the dried broccoli stalk samples (DBS) (Table 1). The DBS showed a similar proximate composition when compared with USDA database results for broccoli stalk [21]. The protein content ranged from 5.6% in DBS to 3.8% in IF_B_, with a significantly lower content in the DF-rich fractions, since the majority of the proteins were removed during the extraction process because of their solubility in ethanol or water. The total carbohydrates (not including DF) were estimated by difference, ranging from 19% in TF_B_ to 44% in DBS. In DBS, the total carbohydrates were significantly higher because the whole sample of broccoli stalk contained starch as a complex polysaccharide and soluble sugars, which were lower in the DF fractions because they were partially removed in the extraction process.

Total dietary fibre (TDF) was the main component of the DF-rich fractions, reaching 68.9% in TF_B_ and 60.8% in IF_B_, being significantly different compared to the DBS (38%). Regarding the classification of the TDF, the content of IDF ranged from 35% in the DBS to 54% in the DF-rich fractions, whereas the SDF content was lower, being 14.7% for TF_B_, 6.8% for IF_B_ and 3.2% for DBS. The contents of IDF and SDF in DBS were similar to those described by Schäfer et al. (2017) [22] for fresh stalks. The content of TDF was highest in TF_B_ (68.9%), since the extraction procedure with ethanol allowed the precipitation of the insoluble and soluble non-starch polysaccharides, SDF representing 21.3% of the TDF. Although IF_B_ was only extracted with hot water, it had a small proportion of SDF, representing 10% of the TDF. Interestingly, this totally green procedure used for IF_B_ extraction, avoiding the use of organic solvents, yielded a similar amount of IDF and half the amount of SDF compared to the TF_B_ process.

The content of ashes was higher in DBS, with a mean value of 12.2%—similar to those reported in the scientific literature, around 10% [21,23]. However, the extraction process to obtain TF_B_ and IF_B_ also removed part of the mineral content, reducing significantly the content of total ash in these fibre-rich fractions to 7.5 and 10.8%, respectively.

Regarding the mineral composition of the samples, potassium and calcium showed the highest contents, while trace elements, such as zinc and iron, had the lowest contents (Table 1). The mineral profile showed a similar proportion for broccoli stalks to that reported by USDA (2019) [21] and Liu et al. (2018) [3]. The results obtained for potassium in DBS samples were between those reported for broccoli and cabbage stalks by other authors ranging from 35 mg/g to 182 mg/g and for calcium were lower than these values raging the data reported between 5.2 and 17 mg/g [3,21,24]. In addition, the two fibre-rich fractions obtained from broccoli stalks had an important content of minerals, reflecting the fact that mineral-binding properties of fibre have been reported in several studies. This binding capacity is affected by several factors, such as the type and amount of fibre, the pH and the ionic strength of the fibre. In this sense, it has been reported that iron has a binding affinity with pectin, and TF_B_ had the highest amount of iron and also the highest proportion of SDF [25]. In addition, zinc has a high binding affinity with respect to insoluble fibres [4,25], and IF_B_ exhibited the highest content of zinc together with the highest proportion of IDF.

The abundance of different monosaccharides (pentoses and hexoses) and uronic acids in the DBS and both fibre fractions (TF_B_ and IF_B_) is shown in Table 2. Of the neutral sugars, glucose was the main monosaccharide in DBS, whereas arabinose and xylose were the main monosaccharides in TF_B_ and IF_B_, respectively. The neutral sugar proportions in DBS were similar to those reported by other authors [22,26], glucose and uronic acids being the main components and rhamnose and fucose the minor ones. In addition, DBS showed a higher content of mannose and lower amounts of uronic acids than the two DF fractions. Since the extraction method led to a difference in the composition of DF between the fractions, the compositions of neutral sugars and uronic acids were also different. Galactose and uronic acids were found in a higher proportion in TF_B_, whereas IF_B_ was mainly composed of rhamnose, fucose, arabinose and xylose. The high content of uronic acids in the DF from broccoli stalks deserves a mention, ranging from 30% for DBS to 49.3% for TF_B_ (Table 2); these results are in agreement with those reported by other authors for different parts of broccoli plants and their fibre residues [10,27]. By contrast, Schäfer et al. (2017) [22] reported that arabinose and glucose were the main components of the insoluble fibre obtained from broccoli stalks by methanolysis and acid hydrolysis, respectively, which shows that the extraction method used to isolate the DF from broccoli stalks determines the final composition of the fibre-rich fractions.

Based on the composition of neutral sugars and uronic acids, the contents of cellulose, hemicellulose and pectin were estimated as indicated in Table 2. Whereas DBS was the sample with the highest proportion of cellulose (18.5%), in TF_B_ and IF_B_ cellulose only represented around 3.5% of TDF. The highest proportion of hemicellulose was found in IF_B_ and DBS, a result of their contents of fucose, xylose, mannose and glucose (Table 2), generally representing a high proportion of IDF. However, TF_B_ showed the highest content of pectin because of its content of rhamnose, arabinose, galactose and uronic acids, being the fraction with the highest proportion of SDF.

To obtain more information about the chemical characteristics of the hemicellulose and pectin structure in the samples, different indexes and sugar ratios were calculated according to Houben et al. (2011) [27] (Table 3). The contribution of mannose to the hemicellulose composition was higher for DBS, indicating a higher solubility in water for this polysaccharide [28]. In contrast, the xylose content was highest in IF_B_ and TF_B_, indicating that hemicellulose was less soluble than the hemicellulose of DBS. Regarding pectin linearity, the results show that pectin molecules in TF_B_ seemed to be more linear than those from IF_B_ and DBS, and that they were also longer in TF_B_ because of the higher content of SDF. The linearity of the pectin chains could provide some interesting physicochemical properties, such as the emulsifying ability, as higher flexibility of the molecules provides better interfacial properties for the stabilisation of emulsions [27,29]. Moreover, the rhamnose contribution was higher in DBS and IF_B_, indicating that the pectin in these samples had the highest degree of branching with lateral chains of galactose and arabinose (RG-I). However, the lateral chains were longer in TF_B_ since it had the highest content of arabinose and galactose. Hence, the different extraction procedures used in the isolation of TF_B_ and IF_B_ led to significant changes in the molecular structure of pectin compared with the original pectins in DBS. As other authors have reported and as mentioned above, the different conditions used in the extraction process, such as temperature, solvents or pH among others, may lead to changes not only in the chemical composition but also in the molecular structure of pectin [30], which may determine the physicochemical and technological properties of the isolated DF-rich fractions.

The physicochemical properties of the DF fractions obtained from broccoli stalk are shown in Table 4, providing information about their potential technological properties as food ingredients and their possible physiological effects in the gastrointestinal system after consumption [31,32]. The hydration properties analysed were the water retention capacity (WRC) and swelling water capacity (SWC), which indicate the capacity of the fibre to retain water in its structure. Higher values of WRC lead to an increase in the faecal volume, facilitating evacuation and reducing the pressure in the rectum, which help to prevent intestinal diseases. Furthermore, when SWC increases the feeling of satiety also increases [33]. IF_B_ was the fraction with the highest values for the hydration properties and TF_B_ showed the lowest values, with significant differences among the samples. In accordance with Belkheiri et al. (2021) [30], who reported that branched chains may negatively affect the gelation properties, in the present study TF_B_ showed the highest branching ratio for pectin and the lowest WRC and SWC, these properties affecting the potential gelling capacity. The hydration properties of our samples were only positively correlated with the hemicellulose content, with r = 0.87 (*p*-value < 0.005) for WRC and r = 0.93 (*p*-value < 0.001) for SWC. However, the higher WRC and SWC of DBS, when compared with TF_B_, could be related also to the retention of water by other components, such as starch, since the content of carbohydrates was highest in this sample (Table 1). The WRC values of the samples from broccoli stalks (3.9–8.2 g of water/g) were similar to that reported for a mixture of broccoli stalks and leaves (8.8 g of water/g) [1] and for the alcohol-insoluble residue from a mixture of broccoli by-products (12.5 g of water/g) [10]. These results are also similar to those for other fibre-rich ingredients derived from tomato peels (6.7 g of water/g) [34] and raspberry (5–10 g of water/g) [31]. However, our values are higher than those reported for some fruit by-products, such as apple pomace and grapefruit peels: 1.6 and 2.3 g of water/g, respectively [30]. The SWC of DBS and IF_B_—17.1 and 20.3 mL water/g, respectively, was higher than the values reported by Rivas et al. (2022) [10], and only TF_B_ showed values similar to those reported by these authors for different DF fractions isolated from broccoli by-products (5–13 mL water/g) [10].

The FAC represents the capacity to retain fat or oil, an attribute that is used in industry to stabilise fatty products and which depends on the chemical and physical structure of the fibre. The highest value was obtained for DBS (4 g of oil/g), being similar to that obtained for broccoli stalk (6.7 g of oil/g) and for an alcohol-insoluble residue from broccoli by-products (4.8 g of oil/g) by other authors [10]. Regarding the osmotic pressure, the lowest value was obtained for TF_B_ (157 mmol/kg) and the highest for DBS (225 mmol/kg), but none of them should induce diarrhoea after consumption as these values are below the physiological values reported previously (290 mmol/kg) [35].

For the TPC (Table 4) measured by spectrophotometric assay (Folin–Ciocalteau method), DBS showed the highest value (154.7 mg/100 g of d.w.), followed by IF_B_ (139 mg/100 g of d.w.) and TF_B_ (39.3 mg/100 g of d.w.), the order being the same as in the HPLC-DAD analysis. Our results for DBS and IF_B_ are similar to the TPC content in broccoli stalks reported by other authors, ranging from 130 to 170 mg GAE/100 g [3,10]. Regarding the antioxidant capacity, which was only analysed in the extractable (poly)phenols (EPP) extract, it is of note that, for both methods, DBS showed the highest antioxidant capacity, followed by IF_B_ and TF_B_. These results are correlated with the content of total (poly)phenolic compounds in the samples, with a correlation coefficient of r = 0.88 (*p* < 0.05). Shi et al. (2019) [1] found an antioxidant capacity of 203.7 µmol TE/g for pomace from broccoli stalks and leaves higher than our values. As previously indicated, other parts of the plant such as florets and leaves may have a higher content of bioactive compounds and, therefore, a higher antioxidant capacity. Moreover, the conditions of the extraction, such as the temperature, solvents and time of maceration, also influence the release of bioactive compounds from the sample.

Shi et al. (2019) [1] also analysed a washed pomace, which was obtained by a process similar to the one used to obtain IF_B_ in our study. In this case, an antioxidant capacity of 7.3 µmol TE/g was observed in the ORAC assay, a value lower than ours, indicating that during the extraction process of that fraction part of the original compounds were lost and, therefore, the antioxidant capacity was reduced [1].

Regarding individual (poly)phenols, the sinapic acid derivatives and chlorogenic acid derivatives groups were the compounds identified by HPLC-DAD in all samples. It is noteworthy that NEPP represented 98%, 97% and 79% of the total (poly)phenols for TF_B_, IF_B_ and DBS, respectively. On the other hand, EPP represented around 21% in DBS and a very small proportion in the DF-rich fractions (Figure 1). 

The total EPP content was significantly higher in DBS than in IF_B_ and TF_B_, with mean values of 10.8, 2.0 and 0.5 mg/100 g d.w., respectively (Figure 1). By contrast, the content of NEPP was highest in IF_B_ (72.7 mg/100 g d.w.), being significantly higher than in DBS (40.5 mg/100 g d.w.) and TF_B_ (28.5 mg/100 g d.w.) (Figure 1). Other authors reported higher values (7–111 mg/g d.w.), perhaps because of the use of different broccoli varieties or growing conditions [36]. These findings show that sinapic acid derivatives and chlorogenic acid derivatives were bound to the fibre compounds as part of the cell wall, as has been reported previously [37,38].

The differences observed in the contents of EPP and NEPP among the DF-rich fractions of broccoli stalk can be explained by the extraction procedure applied for each sample. So, in TF_B_ and IF_B_, part of the EPP compounds were removed after the solubilisation in 80% ethanol and in water, respectively. However, for NEPP, the chlorogenic and sinapic acid derivatives linked to DF compounds were 2.6-fold and 1.8-fold higher in IF_B_ than in DBS and TF_B_, respectively. The small contents of EPP in the samples may explain the lower antioxidant capacities found in this work compared to other reports, as the content of (poly)phenols in a sample is highly correlated to the antioxidant capacity and clearly depends on the extraction method used.

The analysis of glucosinolates was carried out in both the extractable and non-extractable fractions, but no glucosinolates were found in the latter, showing that these bioactive compounds, are stored in organs throughout and they are particularly high in in S-cells, between the endoderms and the phloem cells, and may not be attached to cell wall components [39]. Little information exists about this subject in the literature. Gonzales et al. (2015) [40] also found glucosinolates in the extractable fraction, but not in the non-extractable fraction, in Brussels sprouts [40]. The content of the identified glucosinolates is shown in Figure 2, glucoraphanin (GRA) being the only aliphatic glucosinolate quantified in the three samples, with a mean value around 40 mg/100 g d.w. These results are in agreement with those of other authors, who reported that the major glucosinolate in broccoli is GRA, representing in some cases up to 70% of the total [3,41,42]. The remaining identified glucosinolates belong to the indolic group, with significant differences among the samples as a function of the extraction method. Apart from GRA, 4-methoxyglucobrassicin (MGB) and neoglucobrassicin (NGB) were found in all three samples, whereas glucobrassicin (GBS) was only found in DBS and IF_B_, and 4-hydroxyglucobrassicin (HGB) was only quantified in DBS. 

The total content of glucosinolates in DBS was within the range reported by other authors for broccoli stalk samples, between 69 and 140 mg/100 g d.w. [36,42]. It is worthy of mention that although glucosinolates are soluble in alcohol and water, the solvents used in the extractions of DF-rich fractions, more than 50 and 70% of the total glucosinolates from the raw material remained in the TF_B_ and IF_B_ samples, respectively. This is the first time that the presence of glucosinolates in DF fractions extracted from broccoli by-products has been described. In addition, this reduction was mainly associated with the indolic glucosinolates, since the aliphatic glucosinolate GRA was not removed during the extraction process.

Taking into consideration the physicochemical characterisation of the two DF-rich fractions, we selected IF_B_ as the main fraction of interest because it was extracted following a totally green, environmentally friendly process that is less contaminating because of the use of only hot water without the application of organic solvents. In addition, this process yielded a DF-rich fraction with a similar amount of TDF, better technological properties, a higher content of bioactive compounds and a greater antioxidant capacity, in comparison with TF_B_. For this reason, the prebiotic effect was only evaluated in IF_B_ and was compared with that of DBS. 

Both samples were subjected to an in vitro human faecal fermentation and the SCFAs produced (acetate, propionate, butyrate, isobutyrate, isovalerate, valerate, isocaproate, caproate and heptanoate), after the fermentation of non-starch polysaccharides (NSP) by the gut microbiota, were measured at 0, 4, 8, 24 and 48 h (Figure 3). During this fermentation, a negative control without substrate was also assayed to evaluate the SCFAs produced in the absence of additional substrate. As shown in Figure 3, SCFA production increased during the fermentation of both samples by the microbiota. The two-way ANOVA showed that both factors, sample and time of fermentation, significantly determined the content of all SCFAs, as did the interaction sample x time (*p* < 0.05). As has been reported previously, the main SCFA found was acetate, followed by propionate and butyrate, with an approximate molar ratio of 60:23:17, respectively [43]. The minor SCFAs were represented as the sum of the contents of isobutyrate, valerate, isovalerate, caproate and isocaproate, isobutyrate and isovalerate being the most abundant. 

The fermentation of IF_B_ led to the greatest formation of acetate, some minor SCFAs (isobutyrate. isovalerate and isocaproate) and total SCFAs at 48 h. In contrast, the fermentation of DBS yielded a higher content of butyrate at 48 h than that of IF_B_. In the negative control, without a fibre sample, the content of total SCFAs increased slightly in the first 4 hours of fermentation, reaching a plateau after that with a content that was 75.5% and 78.3% lower than for DBS and IF_B_, respectively. For other minor SCFAs at 24 h of fermentation, the control showed a value up to 4-fold higher compared with the fermentations carried out with the addition of the samples, which could be due to the consumption of tryptone by the bacteria, leading to a higher production of isobutyrate, valerate and isovalerate as end products from this amino acid degradation pathway [44].

A higher butyrate content after fermentation of DBS, compared to IF_B_, could be explained by the higher content of carbohydrates, including starch, in the sample of whole stalks. It is assumed that starch is mainly absorbed in the small intestine, but some fraction of starch can be resistant to digestion, reaching the colon where it can be fermented by microbiota, leading to the formation of butyrate [45]. Even though the content of resistant starch was not analysed in the samples, a small fraction could have remained in the samples after the manipulation and freeze-drying process, since the whole stalk showed a high content of soluble carbohydrates (Table 1).

In general, IF_B_ exhibited a greater prebiotic effect than DBS, leading to a higher content of total SCFAs with significant differences in the production of acetate, butyrate and other minor SCFAs (isobutyrate, isovalerate, valerate, isocaproate, caproate and heptanoate). However, the formation of propionate was similar after the fermentation of both samples. So, the molar distribution and the total content of SCFAs depended on the type and amount of the DF, being determined by the composition of the substrate. Moreover, in general, not only is the chemical composition of fibre important with respect to determining the prebiotic effect of plant foods, but also the physicochemical properties such as the viscosity and hydration properties influence the fermentability. In this regard, pectin is the fraction that is fermented preferentially by the microbiota [46,47,48], but cellulose and hemicellulose could also be used and converted into SCFAs. According to other authors [48], samples with higher contents of uronic acids and cellulose, such as apple and celery stalk, lead to a higher production of total SCFAs during in vitro fermentation compared with other products having an absence of uronic acids and a low content of cellulose, like banana. In addition, the in vitro fermentation of citrus pectin led to a high production of acetate and butyrate [49].

Since the content of uronic acids was higher in IF_B_ than in DBS (Table 2), the formation of total SCFAs was significantly higher in this DF-rich fraction. Moreover, the prebiotic effect of IF_B_ could be attributed to a synergistic effect of the fibre and the bioactive compounds, since IF_B_ showed the highest content of DF and significant amounts of (poly)phenols and glucosinolates, to which has been previously attributed a prebiotic-like effect leading to the modulation of the microbiota [50,51,52].

Pearson´s correlation shows that the contents of pectin and NEPP were significantly and positively correlated (*p* < 0.05) with the production of acetate (r = 0.9 and r = 1), other minor SCFAs (r = 0.9 and r = 1) and total SCFAs (r = 0.9 and r = 0.9) during the in vitro fermentation. On the other hand, the cellulose and EPP contents were correlated negatively with the production of acetate, minor SCFAs and total SCFAs and positively with the production of butyrate, while there was no significant relationship in the case of propionate.

B According to the findings of Rivas et al. (2022) [10], the fibre obtained from broccoli by-products can be considered ingredients with a prebiotic effect, since it increased the populations of *Lactobacillus* at the same time that it increased the production of SCFA. Although the impact of the DF fractions on the microbiota has not been analysed in our study because of the increased of SCFA during in vitro fermentation and based on the definition of prebiotic and prebiotic effect reported by Gibson et al. (2017) [53], it is possible to state the potential prebiotic effect of our sample. 

## 3. Materials and Methods

### 3.1. Samples

Broccoli stalks from broccoli plant cv. Parthenon (*Brassica oleracea*) were provided by the company “Agrícola San Luis” (La Hoya, Murcia, Spain) as an industrial by-product from the broccoli crop cultivated in Caniles, in Mediterranean south-eastern Spain (Granada, lat. 37°26′03″ N, long. 2°43′28″ W). The stalks were selected from the autumn harvest based on a previous seasonal study [54]. The samples were directly processed in the laboratory and stored at −20 °C.

Different samples were obtained from the fresh broccoli stalks and used in this experimental work. First, the stalks were freeze-dried in a Lyoquest freeze-dryer (Telstar, Bensalem, PA, USA) to obtain the dried broccoli stalks (DBS), which were milled to obtain a fine powder. The other two samples were the fibre-rich fractions, total fibre (TF_B_) and insoluble fibre (IF_B_), extracted from fresh broccoli stalks.

### 3.2. Extraction of DF-Rich Fraction

For the extraction of TF_B_, the broccoli stalks were ground in a Thermomix TM-31 and mixed with 80% ethanol, at a ratio of 1/2.5 (*m*/*v*) (Figure 4). The ethanol (80%) was previously heated to 70 °C to facilitate the precipitation of the soluble and insoluble components and to inactivate myrosinase. Each mixture was stirred for 30 min and then centrifuged at 4500× *g* for 5 min; the resulting pellet was left on a convection drying oven at 45 °C until it reached a constant weight. To obtain the IF_B_ fraction, broccoli stalks were mixed with water, previously heated to 70 °C to inactivate myrosinase, at a ratio of 1/2.5 (*m*/*v*) and then stirred for 30 min (Figure 4). Each extract was centrifuged at 4500× *g* for 5 min and the resulting pellet was left on a convection drying oven at 45 °C until it reached a constant weight. The dried samples were crushed separately and stored at −20 °C.

### 3.3. Nutritional Composition

The samples (DBS, TF_B_ and IF_B_) were analysed to determine the proximate composition using the AOAC official methods [55]: moisture (method 964.22), protein by the Kjeldahl method (method 955.03) and ash by incineration (method 923.03). Total carbohydrates were calculated by difference with the other components. Total dietary fibre (TDF), insoluble dietary fibre (IDF) and soluble dietary fibre (SDF) were determined following the enzymatic and gravimetric method described by Prosky et al. (1988) [56], using a Fibertec E 1023 system (Höganäs, Sweden). The mineral composition (calcium, magnesium, sodium, potassium, iron and zinc) was analysed by inductively coupled plasma optical emission spectroscopy, using an ICAP 6500 Duo model (Thermo Fisher Scientific, Waltham, MA, USA), after microwave-assisted digestion in an UltraCLAVE (Milestone, Sorisole, Italy) with H_2_O_2_/HNO_3_ (1/4, *v/v*).

### 3.4. Proportion of Neutral Sugars and Uronic Acids in the DF

For the characterisation of the chemical composition of the DF in the three samples, neutral sugars and uronic acids were analysed. For the analysis of neutral sugars, the GLC-FID method described by Englyst et al. (1992) [57] was followed, after the acid hydrolysis of the DF followed by the derivatisation of neutral sugars to form the alditol acetate derivatives. The analysis was conducted by GLC in an Agilent 7890B model (Agilent, Machelen, Belgium) equipped with a flame ionisation detector. A neutral sugars mix solution (rhamnose, fucose, arabinose, xylose, mannose, galactose and glucose) was used as the source of the standards and β-D-allose (2595-97-3, Thermo Scientific, Madrid, Spain) as the internal standard. The determination of uronic acids was carried out using the colorimetric method described by Scott (1979) [58].

The contents of individual neutral sugars and uronic acids were expressed as percentages (%). The percentage pectin, hemicellulose and cellulose contents in the samples were estimated based on the calculations proposed by Houben et al. (2011) [28] and Umaña et al. (2016) [59]. 

### 3.5. Physicochemical Properties

The water retention capacity (WRC), swelling capacity (SWC), fat absorption capacity (FAC) and osmotic pressure were analysed as physicochemical properties of the samples obtained, following the methodology previously described by Navarro-González et al. [34].

### 3.6. Analysis of EPP and NEPP

To analyse EPP and NEPP, two different extraction procedures described by Arranz et al. (2009) [60] were followed, with some modifications. For EPP, 0.35 g of each sample were mixed with 1 mL of methanol/water/formic acid (79/19/1, *v/v/v*), vortexed for 1 min and centrifuged at 4500× *g* for 10 min at room temperature. The supernatant was evaporated with a Laborota-4002 rotatory evaporator (Heidolph, Schwabach, Germany), and the pellet was resuspended in 10 mL of distilled water. After that, each sample was passed through a previously activated C18-SPE cartridge (Waters Corporation, Milford, MA, USA) and the compounds of interest were recovered in 1 mL of methanol; this was evaporated in a vacuum concentrator (model 5301, Eppendorf, Hamburg, Germany) and the compounds were dissolved in 0.25 mL of methanol/formic acid/acetonitrile (49.5/0.5/50, *v/v/v*).

For NEPP, the residues obtained from the extraction of EPP were resuspended in 5 mL of methanol/H_2_SO_4_ (9/1, *v/v*) and incubated at 85 °C for 20 h. Then, these mixtures were centrifuged at 4500× *g* for 10 min at room temperature and the supernatants were evaporated with a Laborota-4002 rotatory evaporator (Heidolph, Schwabach, Germany). After resuspension in 10 mL of distilled water, the samples were passed through a pre-conditioned C18-SPE cartridge (Waters Corporation, Milford, MA, USA), following the same process described above. 

The (poly)phenols in both fractions, EPP and NEPP, were identified by HPLC in an Agilent 1200 model (Agilent Technologies, Waldbronn, Germany), equipped with a mass detector in series (HPLC-DAD-ESI-MS_n_), following their MS and MS^2^ [M-H]^-^ fragmentation ions, UV-visible spectra and elution order, according to the method described by Francisco et al. (2009) [61]. The quantification was carried out based on the retention times and UV spectra described previously for similar acquisition conditions [62], following the same procedure by using an HPLC with DAD (Agilent 1260 Infinity, Agilent Technologies, Waldbronn, Germany). Chromatograms were recorded at 320 nm using chlorogenic acid (Sigma-Aldrich Chemie GmbH, Steinheim, Germany) and sinapinic acid (Sigma, St. Louis, MO, USA) as standards for chlorogenic and sinapic acid derivatives, respectively. The results were expressed as mg/100 g dry weight (d.w.).

### 3.7. Identification and Quantification of Glucosinolates

Glucosinolates were analysed in the same extract as the EEP and NEPP. For the identification and quantification of aliphatic and indolic glucosinolates the HPLC methods described above for the analysis of (poly)phenols were followed [54,55]. However, chromatograms were registered at 227 nm, using sinigrin and glucobrassicin (Phytoplan, Heidelberg, Germany) as standards for aliphatic and indolic glucosinolates, respectively. The content was expressed as mg/100 g d.w.

### 3.8. Antioxidant Capacity

Total phenolic compounds were analysed using Folin–Ciocalteu’s colorimetric assay as described by Singleton and Rossi (1965) [63]. For the colorimetric assay a microplate spectrophotometer (BioTek Instruments, Winooski, VT, USA) was used. Gallic acid (Riedel-de Haën, Hannover, Germany) was used as the standard, and the total phenolic content in the samples was expressed as mg of gallic acid equivalents (GAE)/100 g d.w.

The antioxidant capacity of the EPP extracts was analysed by two different methods: the ferric reducing antioxidant power (FRAP) and the oxygen radical absorbance capacity (ORAC), following the procedures described by Benzie and Strain (1996) [64] and Ou et al. (2001) [65], respectively. The methods were adapted to a microplate spectrophotometer (BioTek Instruments, Winooski, VT, USA) and the results were expressed as µmol of TE/g d.w.

### 3.9. In Vitro Human Faecal Fermentation

Previous to the in vitro fermentations, samples were subjected to simulated gastrointestinal digestion, following the in vitro method from the INFOGEST protocol [66]. After that, in vitro fermentations were carried out according to the method described by González-Barrio et al. (2011) [67]. Human faecal samples were collected from nine healthy normal-weight women, aged 20 to 28 years old. The inclusion criteria were that each individual was a non-smoker with stable food habits, who did not present any symptoms of gastrointestinal disease, had not taken antibiotics for at least 4–6 months before the study, did not follow any dietary restrictions and did not take any food supplements, prebiotics or probiotics. Moreover, the volunteers had to follow a pre-established diet for two days prior to the study, mainly to remove the main sources of fibre, (poly)phenols and glucosinolates. This study was approved by the Research Ethics Commission (2664/2019) of the University of Murcia. Fresh faeces were collected in a flask containing an AnaeroGen^TM^ Sacket (AN35, Oxoid^®^, Hampshire, UK), to produce anaerobic conditions and avoid microbial modifications, and were processed in the following 2 h. The faecal samples were pooled to increase the microbial diversity. Samples of fresh faeces were homogenised with phosphate buffer to obtain 32% faecal suspensions. Then, 5 mL of faecal suspension were added to 44 mL of fermentation medium at pH 7 and placed in a 100 mL McCartney bottle, and 0.5 g (1%) of each sample of extracted plant material were added, after being dissolved in 1 mL of water, to each fermentation bottle. A negative control without any sample added was also incubated. After this, the fermentation bottles were purged with nitrogen and then incubated for 48 h at 37 °C in a shaking bath, simulating colonic lumen conditions. Aliquots of the fermented faecal samples were collected at the baseline (0 h) and after 4, 8, 24 and 48 h and immediately stored at −80 °C prior to analysis of the SCFAs.

### 3.10. Analysis of SCFAs after in vitro Fermentation by GCL

The SCFAs were analysed according to the protocol described by Baenas et al. (2020) [31]. Briefly, the fermented faecal sample was centrifuged and the supernatant obtained mixed with 0.65 mL of a solution of 20% formic acid/methanol/2-ethyl butyric acid (internal standard, 2 mg/mL in methanol) (1/4.5/1, *v/v/v*). After that, the sample was centrifuged, filtered and analysed by GC-FID. Chromatographic analysis was carried out using an Agilent 7890A GC system equipped with a flame ionisation detector (FID) and a 7683B automatic injector (Agilent Technologies, Santa Clara, CA, USA). A fused-silica capillary column (Nukol TM, Supelco, St. Louis, MO, USA) of 30 m × 0.25 mm, i.d. 0.25 µm coated, was used to separate the SCFAs. Helium was supplied as the carrier gas at a flow rate of 25 mL/min. The initial oven temperature was 80 °C and it was kept constant for 5 min and then raised to 185 °C at a rate of 5 °C/min. Samples (2 µL) were injected in splitless mode, with an injection port temperature of 220 °C. The flow rates of hydrogen and air were 30 and 400 mL/min, respectively. The temperature of the FID was 220 °C and the running time for each analysis was 26 min.

The SCFAs were identified by comparison with the retention times of authentic standards (Supelco, St. Louis, MO, USA). Quantification was based on calibration curves constructed for a set of SCFAs standards (acetic acid, propionic acid, isobutyric acid, butyric acid, isovaleric acid, valeric acid, isocaproic acid, caproic acid and heptanoic acid) and the results were expressed as mM.

### 3.11. Statistical Analysis

The statistical analysis was carried out using R studio, version 4.0.5 (R Foundation for Statistical Computing, Vienna, Austria). Normality was determined by the Shapiro−Wilk test. The homogeneity of variances was analysed using the Bartlett test. One-way analysis of variance (ANOVA) was performed to reveal significant differences among the samples for the parameters analysed. Tukey´s test was used as a post hoc test. Moreover, two-way ANOVA was performed to determine the prebiotic effect of the samples, considering the effect of the different samples and the time of fermentation in the SCFAs production during the in vitro fermentation analysis. Correlation analysis was performed for the relationships between the physicochemical properties and fibre composition parameters, and among the parameters of the composition of the DF of the samples (DBS and IF_B_) and the SCFAs produced during fermentation. Differences were considered significant at a *p*-value < 0.05.

## 4. Conclusions

Fibre-rich fractions obtained from broccoli stalks can be considered interesting from a nutritional point of view because of the presence of fibre, (poly)phenols and glucosinolates. In addition, IF_B_ showed a good prebiotic effect, resulting in the production of SCFAs; this, together with its content of bioactive compounds and fibre, makes it an interesting fraction at both the nutritional and technological levels. Besides, the extraction method employed for IF_B_, using water, is a clean and environmentally friendly way of recycling an industrial by-product as a new ingredient in the food and/or pharmaceutical industry. The final physicochemical properties of such DF-rich ingredients depend not only on the proportion of soluble or insoluble fibre, but also on the extraction method, allowing the aqueous method to obtain ingredients with improved hydration properties.

## Figures and Tables

**Figure 1 ijms-23-13309-f001:**
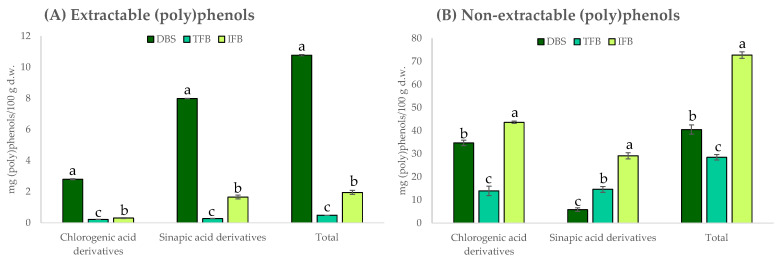
Contents of individual (poly)phenols (chlorogenic acid derivatives and sinapic acid derivatives) and the total contents of (poly)phenols determined as the sum of the individual compounds (mg/100 g d.w.): (**A**) extractable phenolics (EPP) and (**B**) non-extractable phenolics (NEPP), analysed by HPLC-DAD for freeze-dried broccoli stalk (■DBS), the total fibre fraction (■TF_B_) and the insoluble fibre fraction (■IF_B_). Values are expressed as mean ± SD (n = 3). Different letters (a–c) indicate significant differences among the samples (*p* < 0.05) for each individual group and the total (poly)phenols.

**Figure 2 ijms-23-13309-f002:**
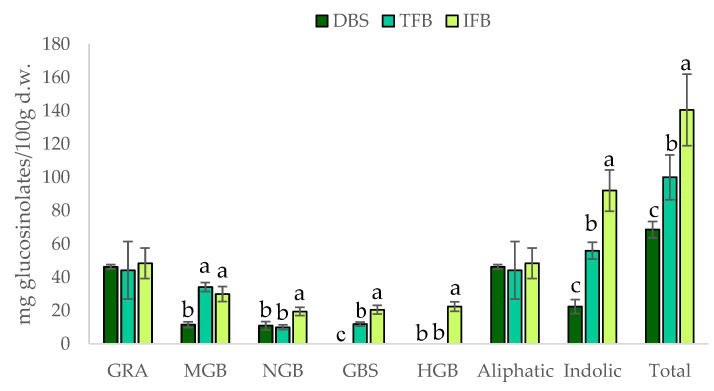
Contents of glucosinolates (mg/100 g of d.w.), analysed by HPLC-DAD, of freeze-dried broccoli stalk (■DBS), the total fibre fraction (■TF_B_) and the insoluble fibre fraction (■IF_B_). Glucosinolates are presented as individual compounds (GRA: glucoraphanin (4-methylsulphinybutyl-gls), MGB: 4-methoxyglucobrassicin (4-methoxy-3-indolylmethyl-gls), NGB: neoglucobrassicin (1-methoxy-3-indolylmethyl-gls), GBS: glucobrassicin (3-indolylmethyl-gls), HGB: 4-hydroxyglucobrassicin (4-hydroxy-3-indolylmethyl-gls)). Values are expressed as mean ± SD (n = 3). Different letters (a–c) indicate significant differences (*p* < 0.05) among the samples for each individual glucosinolate and the total glucosinolates.

**Figure 3 ijms-23-13309-f003:**
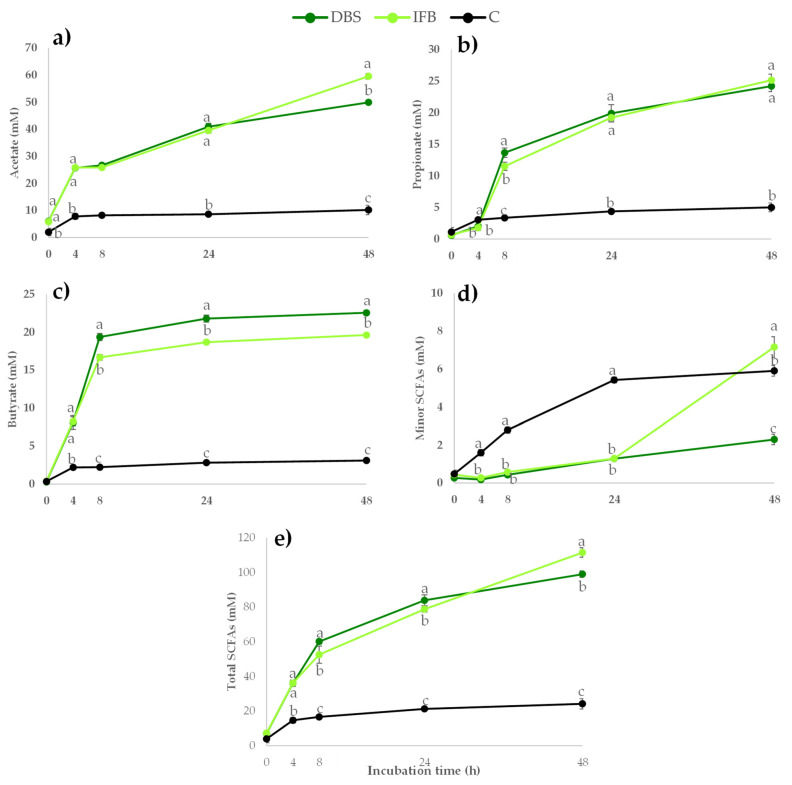
SCFAs production (acetate (**a**), propionate (**b**), butyrate (**c**), other minor SCFAs (isobutyrate, isovalerate, valerate, isocaproate, caproate and heptanoate) (**d**) and total SCFAs (**e**)) (mM) during in vitro fermentation of samples obtained from broccoli stalk. Freeze-dried broccoli stalk (●DBS); the insoluble fibre fraction (●IF_B_); the control (●C). Values are expressed as mean ± SD (n = 3). Different letters (a–c) indicate significant differences (*p* < 0.05) among the samples at the same incubation time.

**Figure 4 ijms-23-13309-f004:**
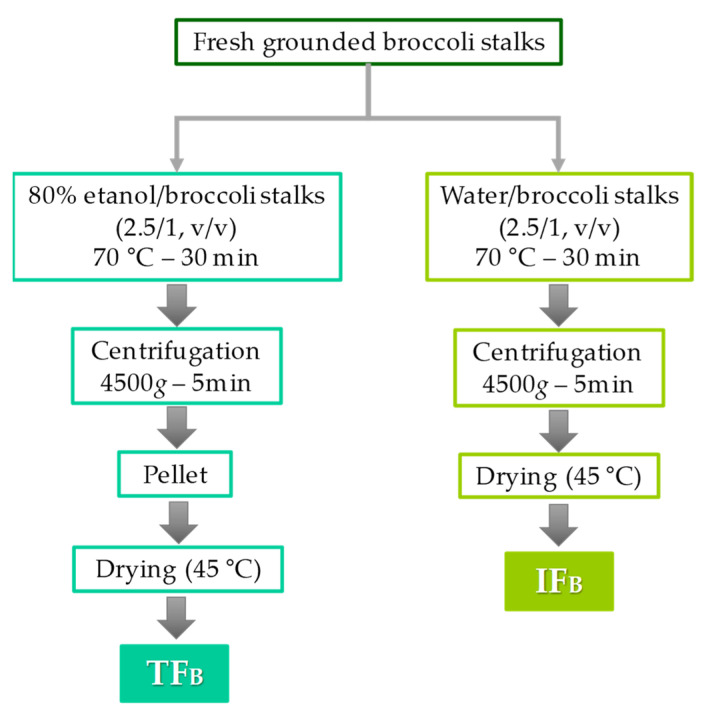
Flow diagram of the procedure used to obtain the fibre-rich fractions, total fibre fraction (TF_B_) and insoluble fibre fraction (IF_B_).

**Table 1 ijms-23-13309-t001:** Nutritional composition, expressed as percentage or g/100 g d.w., and mineral composition (mg/kg) of different samples obtained from broccoli stalk *.

Parameters	DBS	TF_B_	IF_B_
Moisture	0.3 ± 0.0 ^a^	0.2 ± 0.0 ^b^	0.2 ± 0.0 ^b^
Protein	5.6 ± 0.2 ^a^	4.0 ± 0.8 ^b^	3.8 ± 0.0 ^b^
Total carbohydrates **	43.9 ± 0.4 ^a^	19.3 ± 1.2 ^c^	24.4 ± 0.7 ^b^
Total dietary fibre (TDF)	38.0 ± 0.4 ^c^	68.9 ± 0.4 ^a^	60.8 ± 0.4 ^b^
Insoluble dietary fibre (IDF)	34.9 ± 0.1 ^b^	54.3 ± 0.4 ^a^	54.0 ± 0.1 ^a^
Soluble dietary fibre (SDF)	3.2 ± 0.5 ^c^	14.7 ± 0.1 ^a^	6.8 ± 1.0 ^b^
Total ash	12.2 ± 0.2 ^a^	7.5 ± 0.0 ^c^	10.8 ± 0.2 ^b^
K	47365.8 ± 371.7 ^a^	14540.0 ± 294.8 ^c^	31724.1 ± 1613.2 ^b^
Ca	4887.8 ± 56.4 ^b^	6472.4 ± 308.9 ^a^	6234.7 ± 331.9 ^a^
P	4118.5 ± 31.2 ^a^	4138.5 ± 170.4 ^a^	2650.5 ± 145.3 ^b^
Na	2920.1 ± 34.2 ^b^	887.1 ± 56.7 ^c^	8166.1 ± 464.5 ^a^
Mg	2555.2 ± 39.2 ^a^	1332.7 ± 73.6 ^b^	2629.3 ± 148.9 ^a^
Mn	39.6 ± 0.6 ^a^	24.4 ± 1.2 ^b^	39.9 ± 1.4 ^a^
Zn	23.1 ± 0.8 ^b^	14.6 ± 0.7 ^c^	27.9 ± 2.9 ^a^
Fe	15.7 ± 0.8 ^c^	25.0 ± 1.8 ^a^	18.9 ± 1.4 ^b^

* Values are expressed as mean ± SD (n = 3). Different letters (a–c) indicate significant differences (*p* < 0.05) among the samples. ** Carbohydrates were calculated as the difference. Include soluble sugars and starch. Freeze-dried broccoli stalk (DBS); total fibre fraction (TF_B_); insoluble fibre fraction (IF_B_).

**Table 2 ijms-23-13309-t002:** Proportions of neutral sugars, uronic acids, cellulose, hemicellulose and pectin in different samples obtained from broccoli stalk, expressed as percentages (%) *.

% Neutral Sugars	DBS	TF_B_	IF_B_
Rhamnose	1.8 ± 0.1 ^b^	1.7 ± 0.1 ^b^	2.3 ± 0.4 ^a^
Fucose	0.8 ± 0.0 ^b^	0.7 ± 0.1 ^b^	1.1 ± 0.1 ^a^
Arabinose	15.9 ± 0.6 ^b^	16.7 ± 0.8 ^b^	18.5 ± 0.5 ^a^
Xylose	16.4 ± 0.5 ^b^	14.3 ± 0.8 ^c^	20.2 ± 0.6 ^a^
Mannose	4.5 ± 0.1 ^a^	2.4 ± 0.2 ^c^	2.7 ± 0.1 ^b^
Galactose	10.0 ± 0.3 ^b^	11.4 ± 0.2 ^a^	10.7 ± 0.7 ^ab^
Glucose	20.6 ± 2.3 ^a^	3.6 ± 0.3 ^b^	4.0 ± 0.3 ^b^
Uronic acids	30.1 ± 2.8 ^c^	49.3 ± 1.6 ^a^	40.6 ± 1.0 ^b^
% NSP **			
Cellulose ^1^	18.5 ± 2.1 ^a^	3.3 ± 0.2 ^b^	3.6 ± 0.3 ^b^
Hemicellulose ^2^	24.1 ± 0.3 ^a^	17.6 ± 1.2 ^b^	24.4 ± 0.6 ^a^
Pectin ^3^	57.8 ± 2.8 ^c^	79.1 ± 1.2 ^a^	72.1 ± 0.7 ^b^

* Values are expressed as mean ± SD (n = 4). Different letters (a–c) indicate significant differences (*p* < 0.05) among the samples. ** NSP: non-starch polysaccharides. Freeze-dried broccoli stalk (DBS); total fibre fraction (TF_B_); insoluble fibre fraction (IF_B_). ^1^ Cellulose: glucose x 0.9; ^2^ hemicellulose: (fucose + xylose + mannose + (glucose × 0.1)); ^3^ pectin: (rhamnose + arabinose + galactose + uronic acids).

**Table 3 ijms-23-13309-t003:** Sugar ratios for characterisation of pectin and hemicellulose from different samples obtained from broccoli stalk *.

Ratios	DBS	TF_B_	IF_B_
Man contribution ^1^	0.3 ± 0.0 ^a^	0.2 ± 0.0 ^b^	0.1 ± 0.0 ^c^
Linearity of pectins ^2^	0.7 ± 0.1 ^c^	1.1 ± 0.1 ^a^	0.8 ± 0.0 ^b^
Rha/Uro contribution ^3^	0.1 ± 0.0 ^a^	0.03 ± 0.00 ^b^	0.1 ± 0.0 ^a^
RG-I branching ^4^	14.7± 1.0 ^ab^	16.7 ± 1.3 ^a^	12.9 ± 1.9 ^b^

* Values are expressed as mean ± SD (n = 3). Different letters (a–c) indicate significant differences (*p* < 0.05) among the samples. Freeze-dried broccoli stalk (DBS); total fibre fraction (TF_B_); insoluble fibre fraction (IF_B_). ^1^ Contribution of mannans to hemicelluloses: mannose/xylose; ^2^ linearity of pectins: uronic acids/(fucose + rhamnose + arabinose + galactose + xylose); ^3^ contribution of rhamnose and uronic acids to pectins: rhamnose/uronic acids; ^4^ branching of RG-I: (arabinose + galactose)/rhamnose.

**Table 4 ijms-23-13309-t004:** Physicochemical properties, content of total phenolic compounds and antioxidant capacity of samples obtained from broccoli stalk *.

Physicochemical Properties	DBS	TF_B_	IF_B_
Water retention capacity (g of water/g)	6.4 ± 0.9 ^b^	3.9 ± 0.3 ^c^	8.2 ± 0.8 ^a^
Swelling capacity (mL of water/g)	17.1 ± 0.8 ^b^	10.2 ± 0.8 ^c^	20.3 ± 0.8 ^a^
Fat absorption capacity (g of oil/g)	4.0 ± 0.0 ^a^	3.7 ± 0.0 ^b^	2.6 ± 0.1 ^c^
Osmotic pressure (mmol/kg)	225.0 ± 1.0 ^a^	157.0 ± 1.0 ^c^	187.3 ± 3.8 ^b^
Total phenolic compounds(mg GAE/100 g of d.w.)	154.7 ± 8.3 ^a^	39.3 ± 2.4 ^c^	139.0 ± 8.7 ^b^
FRAP (µmol TE/100 g of d.w.)	264.0 ± 9.6 ^a^	102.7 ± 2.1 ^c^	229.6 ± 10.4 ^b^
ORAC (µmol TE/100 g of d.w.)	2821.7 ± 96.5 ^a^	1666.9 ± 40.8 ^c^	1856.4 ± 19.0 ^b^

* Values are expressed as mean ± SD (n = 3). Different letters (a–c) indicate significant differences (*p* < 0.05) among the samples. Freeze-dried broccoli stalk (DBS); total fibre fraction (TF_B_); insoluble fibre fraction (IF_B_).

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
