# Peer review of "Dietary-Fibre-Rich Fractions Isolated from Broccoli Stalks as a Potential Functional Ingredient with Phenolic Compounds and Glucosinolates"

_ijms, 2022, doi:10.3390/ijms232113309_

Round 1
Reviewer 1 Report
Abstract - an abbreviation is used without the full name (short chain fatty acids). Please use the full Latin name for broccoli.
Please avoid repeating words from the title in the keywords. I would suggest restructuring the paragraphs in the introduction to create a more logical sequence of sentences. Please discuss the results of the mineral amount determined with literature references.
It is common to express macroelements in % and microelements in mg/kg when based on dry matter.
Tables - I suggest to present the data more clearly without repeating the units of measurement
Materials and methods - it would be interesting to add more information about the origin (cultivation) of the broccoli "raw material"
Conclusions - please add some recommendations... (The final physicochemical properties of such DF-rich ingredients depend not only on the proportion of soluble or insoluble dietary fibre, but also on the extraction method ....)
Please make sure that the references are written in a consistent way (upper and lower case letters, page numbers, ...)
Author Response
RESPONSE TO REVIEWER 1
Abstract - an abbreviation is used without the full name (short chain fatty acids). The full name is already included.
Please use the full Latin name for broccoli. It is already included in both, the abstract and the introduction, as well in the material and method section.
Please avoid repeating words from the title in the keywords. The keyword “dietary fibre” has been removed.
I would suggest restructuring the paragraphs in the introduction to create a more logical sequence of sentences. This section has been reorganized and rewritten.
Please discuss the results of the mineral amount determined with literature references. We have rewritten the discussion related to the content of minerals, but to our knowledge there are no studies in the scientific literature about the mineral content of broccoli stalk (lines 160-165).
It is common to express macroelements in % and microelements in mg/kg when based on dry matter. The units of the microelements have been changed as is shown in modified Table 1.
Tables - I suggest to present the data more clearly without repeating the units of measurement. Units have been changed and are no longer repeated from the legend.
Materials and methods - it would be interesting to add more information about the origin (cultivation) of the broccoli "raw material". The information about the variety and origin of the crop has been included in section 3.1 (lines 452-456).
Conclusions - please add some recommendations... (The final physicochemical properties of such DF-rich ingredients depend not only on the proportion of soluble or insoluble dietary fibre, but also on the extraction method ....). Explanation has been included in the final paragraph of the conclusion.
Please make sure that the references are written in a consistent way (upper and lower case letters, page numbers, ...). All references have been checked.

Reviewer 2 Report
Dear authors,
Manuscript ijms-1981132 entiteled "Chemical composition, bioactive compounds and functional properties of dietary-fibre-rich fractions obtained from broccoli stalks" and authored by VANESA NUÑEZ GOMEZ , Rocío González-Barrio , Nieves Baenas , Diego A. Moreno , Mª Jesús Periago targets a hot topic and could be potentially of huge interest to the journal readers. Unfortunately there major and minor issues that authors should address before the manuscript can be suggested for publication:
Minor issues:
1. The title have to be changed to a new title that describe major findings opf the work and that encourage the journal readers to read the paper.
2. The introduction section have to be improved : The authors have to describe benefits of SCFAs: this is the major finding of the paper and there is no description of this.
3. The introduction section: please detail the importance, relevance and function of minerals , bioactive compounds (glucosinolates, (poly)phenols and carotenoids) and vitamins in the context of the study. The readers need to be updated about their improtance
4. The conclusion section : please rathar than citing general considerations apply findings of your paper to the results of your work. When you state "The final physicochemical properties of such DF-rich ingredients depend not only on the proportion of soluble or insoluble fibre, but also on the extraction method, which determines the physical structure of the fibre molecules in the extract." please detail your major findings which extraction techniques and which physico-chemical proterties. Please be precise this doesn't help the reader to access your main findings.
Major issue:
The authors state at the end of the manuscript : "Even though the impact of the DF fractions on the microbiota has not been analysed 408 in our study, based on a similar increased production of SCFAs reported by Rivas et al. 409 (2022) [16], who also used broccoli by-products, it is likely that the positive effect observed 410 by these authors, with increased populations of Lactobacillus, also occurred with our sam-411 ples during in vitro fermentations." I am sorry this is speculation this is not science. Therefore I suggest removing prebiotic effect throughpout the manucript and replacing it by " higher content of total SCFAs" which is factual and scientifically sound.
I am looking toward reading an improved version of the manuscript that mainly discard specualtion and that highlight the nice findings without misleasding the readers that I can recommend for publication.
Best regards
Author Response
RESPONSE TO REVIEWER 2
Minor issues:
- The title have to be changed to a new title that describe major findings of the work and that encourage the journal readers to read the paper. We change the original title
- The introduction section have to be improved: The authors have to describe benefits of SCFAs: this is the major finding of the paper and there is no description of this. This section has been improved by including more benefits from the production of SCFAs, including also new references (lines 69-82).
- The introduction section: please detail the importance, relevance and function of minerals, bioactive compounds (glucosinolates, (poly)phenols and carotenoids) and vitamins in the context of the study. The readers need to be updated about their importance. The relevance of these compounds to the study has been included in the second paragraph (lines 45-67).
- The conclusion section: please rather than citing general considerations apply findings of your paper to the results of your work. When you state "The final physicochemical properties of such DF-rich ingredients depend not only on the proportion of soluble or insoluble fibre, but also on the extraction method, which determines the physical structure of the fibre molecules in the extract." please detail your major findings which extraction techniques and which physico-chemical proterties. Please be precise this doesn't help the reader to access your main findings. Explanation has been included in the final paragraph of the conclusion.
Major issue:
The authors state at the end of the manuscript : "Even though the impact of the DF fractions on the microbiota has not been analysed 408 in our study, based on a similar increased production of SCFAs reported by Rivas et al. 409 (2022) [16], who also used broccoli by-products, it is likely that the positive effect observed 410 by these authors, with increased populations of Lactobacillus, also occurred with our sam-411 ples during in vitro fermentations." I am sorry this is speculation this is not science. Therefore, I suggest removing prebiotic effect throughpout the manucript and replacing it by " higher content of total SCFAs" which is factual and scientifically sound. We have rewritten this paragraph, describing the findings of the Rivas et al. (2022). Instead, in relation to the term prebiotic effect, we have included a new reference supporting the fact that it can be considered a prebiotic effect even if it has not been demonstrated by other analyses that the ingredients themselves are prebiotic (lines 440-446).

Round 2
Reviewer 2 Report
Thanks for addressing my comments